# Visfatin and Retinol Binding Protein-4 in Young-Onset Type 2 Diabetes Mellitus

**DOI:** 10.3390/medicina59071278

**Published:** 2023-07-09

**Authors:** Ya-Li Huang, Yen-Lin Chen, Jiunn-Diann Lin, Dee Pei, Pietro Pitrone, Jin-Shuen Chen, Chung-Ze Wu

**Affiliations:** 1Department of Public Health, School of Medicine, College of Medicine, Taipei Medical University, Taipei City 11031, Taiwan; ylhuang@tmu.edu.tw; 2School of Public Health, College of Public Health, Taipei Medical University, Taipei City 11031, Taiwan; 3Department of Pathology, Tri-Service General Hospital, National Defense Medical Center, Taipei City 11490, Taiwan; anthonypatho@gmail.com; 4Division of Endocrinology and Metabolism, Department of Internal Medicine, School of Medicine, College of Medicine, Taipei Medical University, Taipei City 11031, Taiwan; jdlin1971@yahoo.com.tw; 5Division of Endocrinology and Metabolism, Department of Internal Medicine, Shuang Ho Hospital, Taipei Medical University, New Taipei City 23561, Taiwan; 6School of Medicine, College of Medicine, Fu Jen Catholic University, New Taipei City 24352, Taiwan; peidee@gmail.com; 7Division of Endocrinology and Metabolism, Department of Internal Medicine, Fu Jen Catholic University Hospital, New Taipei City 24352, Taiwan; 8Radiology Department, Papardo Hospital, 98100 Messina, Italy; pieropitrone@live.it; 9Deputy Superintendent, Kaohsiung Veterans General Hospital, Kaohsiung City 81362, Taiwan; dgschen@vghks.gov.tw; 10Institute of Precision Medicine, National Sun Yat-sen University, Kaohsiung City 80424, Taiwan; 11Division of Nephrology, Department of Medicine, Tri-Service General Hospital, Taipei City 11490, Taiwan

**Keywords:** young diabetes mellitus, visfatin, retinol binding protein 4, metabolic syndrome, old diabetes mellitus

## Abstract

*Background and Objectives*: The prevalence of type 2 diabetes mellitus in adolescents has increased rapidly in recent decades. However, the role of adipokines on pathophysiology in young-onset type 2 diabetes mellitus (YDM) is not clear. In this article, we explored the relationships between the adipokines (visfatin and retinol binding protein 4 (RBP4)) and metabolic syndrome (MetS) components in both YDM and late-onset type 2 diabetes mellitus (ODM). *Materials and Methods*: There were 36 patients with YDM (23.6 ± 4.8 years) and 36 patients with ODM (54.3 ± 10.1 years) enrolled. Visfatin, RBP4, and MetS components were measured. The relationships between visfatin, RBP4 and MetS components were assessed in YDM and ODM. *Results*: The visfatin, but not the RPB4 level, was significantly higher in YDM than in ODM. After adjusting for age and body mass index, visfatin was not related to any MetS components except that there was a negative correlation with fasting plasma glucose (FPG). As for RPB4, triglyceride was found to be positively and FPG negatively related to RBP4 in YDM. However, in ODM, the only positive relationship that existed was between RBP4 and diastolic blood pressure. *Conclusions:* In conclusion, both visfatin and RBP4 had certain roles in diabetes and MetS although their relationships were different in YDM and ODM. Further studies are needed to explore their physiological and pathological effects in glucose metabolism.

## 1. Introduction

The prevalence of young-onset type 2 diabetes mellitus (young diabetes mellitus; YDM) has been rising dramatically in recent decades worldwide. This is probably due to the parallel increase in the prevalence of obesity in children [1]. Thus, type 2 diabetes mellitus (T2D) has already become the most predominant type of diabetes in adolescents and children in Taiwan [2]. It is well known that the risks of macrovascular and microvascular complications increase with both the duration of T2D and status of glycemic control. However, according to Yeung et al., the glycemic control in YDM is significantly poorer compared with in those with late-onset type 2 diabetes mellitus (old diabetes mellitus; ODM) in Asia [3]. Consequently, it could be postulated that those with YDM are predisposed to have more severe complications than those with?? ODM when they are in their middle age because of the duration and poor control of the diseases. This may have an adverse impact on national productivity and public health. Even though it is important, till now, the pathophysiology of YDM is still incompletely understood.

An important feature of obesity is an accumulation of excessive adipose tissues in subcutaneous and visceral regions. In recent years, adipose tissue is regarded as an endocrine organ that secretes a variety of cytokines regulating metabolism, reproduction, inflammation, and cardiovascular function [4]. Pre-B cell colony-enhancing factor, known as visfatin, is released extracellularly by a variety of cells, including visceral adipocytes and infiltrating macrophages of visceral adipose tissue. It is found to be highly correlated with visceral adiposity [5,6,7]. Although there is still a discrepancy in the relationships between visfatin, glucose metabolism, and lipid profiles, the results of the meta-analysis still show that an elevated visfatin level is higher in subjects with obesity, T2D, metabolic syndrome (MetS), and cardiovascular disease [8]. Furthermore, some research suggested elevated visfatin levels in individuals with insulin resistance-related diseases, such as polycystic ovary syndrome and fatty liver disease [9,10]. This increase appears to be regulated through the signaling pathways of the peroxisome proliferator-activated receptor-γ (PPAR-γ) [11]. Additionally, excessive visfatin levels might induce inflammation and fibrosis in hepatocytes [12]. Consequently, visfatin could potentially serve as a biomarker for the presence of excess visceral fat, inflammation, and endothelial dysfunction [13].

At the same time, retinol binding protein 4 (RBP4) is highly expressed in adipose tissue and found to be higher in subjects with obesity and T2D [14]. This might be due to the finding that it is positively associated with insulin resistance (IR). Several studies have discovered that elevated RBP4 levels are linked to obesity, metabolic syndrome, and an increased risk of cardiovascular disease, fatty liver, and polycystic ovary syndrome [15,16,17,18]. Moreover, RBP4 is associated with T2D-related complications [19], which contribute to the development of atherosclerosis in T2D by regulating the JAK2/STAT3 signaling pathway [20]. However, the *Atherosclerosis Risk in Communities Study* (ARIC Study) found this association between RBP4 and the onset of T2D only in women, not in men [21]. Similar to visfatin, the roles of RBP4 in T2D and IR still remain controversial.

The understanding of the different pathogenesis between YDM and ODM is crucial for reference of preventive strategies on YDM in the future. So far, there is no study investigating the roles of visfatin and RBP4 between YDM and ODM. Comparing their relationships with MetS components between YDM and ODM becomes important. In the present study, we tried to explore and compare the circulating visfatin and RBP4 levels between YDM and ODM, and their relationships with MetS components.

## 2. Materials and Methods

### 2.1. Subjects

We enrolled patients with T2D who visited the outpatient clinic of a local hospital in Taipei, Taiwan. The diagnostic criteria of T2D were based on the ADA guideline (fasting plasma glucose ≥ 7 mmol/dL). All subjects were being treated with only oral hypoglycemic agents at the time of enrollment. Subjects who received insulin injection or experience diabetic ketoacidosis or the presence of major cardiovascular, respiratory, renal, or endocrine disorders were excluded. Subjects who took any medication, except for oral hypoglycemic agents, known to affect the metabolism of glucose and lipid were also excluded. The dose of oral hypoglycemic agents was maintained at least three months prior to the study. In addition, subjects who refused to participate or had missing or incomplete data were excluded. The definition of the YDM group was under 22 years old at the time they were enrolled in the study. On the other hand, for the ODM group, it was more than 40 years old. For patients in the ODM group, their annual health exam or medical records were reviewed for assurance of T2D onset age of more than 40 years. The flowchart of participant selection is displayed in Figure 1. The study was approved by the Institutional Review Board of our institution (approval number: CRC-03-09-01), and the nature, purpose, and potential risks of the study were explained to the subjects before obtaining their consent to participate.

### 2.2. Anthropometry and Laboratory Measurement

Subjects received a stable diet three days before they entered the study. After overnight fasting, subjects visited our clinic at 8 am. Information on medical history was obtained through interviews by the senior nursing staff. A complete physical examination was done and the body mass index (BMI) was calculated as body weight/body height^2^ (kg/m^2^). Waist circumference (WC) was taken at the midway point between the inferior margin of the last rib and crest of the ilium in a horizontal plane and measured to the nearest 0.1 cm. Systolic blood pressure (SBP) and diastolic blood pressure (DBP) were measured by nursing staff using standard mercury sphygmomanometers on the right arm of seated participants who had rested for 5 min. Blood samples were drawn from an antecubital vein for biochemistry analysis.

Plasma was separated from the blood within one hour and stored at −30 °C until analyzed. Circulating visfatin was measured by commercial enzyme immunoassay kits (Phoenix Pharmaceuticals, Belmont, CA, USA). The intra-assay and inter-assay coefficients of variation were <10% and <15%, respectively, for visfatin. RBP4 levels were determined by the Human Retinol-Binding Protein 4 ELISA Kit (Assaypro, MO, USA). The intra-assay and inter-assay coefficients of variation were 3.8% and 9.5%, respectively, for RBP4. Fasting plasm glucose (FPG) was measured by a glucose oxidase method and analyzer (YSI 203 glucose analyzer, Scientific Division, Yellow Spring Instruments, Yellow Spring, OH, USA). Hemoglobulin A1c (HbA1c) was measured by a Bio-Rad Variant II automatic analyzer (BioRad Diagnostic Group, Los Angeles, CA, USA). Triglyceride (TG) was measured using the dry, multilayer analytical slide method in the Fuji Dri-Chem 3000 analyzer (Fuji Photo Film, Minato-Ku, Tokyo, Japan). HDL-C concentration was analyzed using an enzymatic, cholesterol assay method following dextran sulfate precipitation.

### 2.3. Statistical Analysis

Statistical analysis was performed by using SPSS (version 18.0 statistical package for Windows; SPSS, Chicago, IL, USA). Continuous variables were expressed as the mean ± standard deviation. An independent Student *t*-test was used to evaluate visfatin, RBP4, HgbA1c, and MetS components between subjects with YDM or ODM. Because RBP4 and visfatin are tightly related to adiposity, the relationships of RBP4, visfatin, and MetS components were assessed by Pearson’s correlation analysis after adjusting for age and BMI. For further analysis, visfatin and RBP4 were analyzed by multivariate linear regression with age, BMI, and all of the MetS components. All statistical tests were two-sided, and *p* values < 0.05 were considered to indicate statistical significance.

## 3. Results

### 3.1. Comparison Circulating Visfatin, RBP4 and MetS Components between YDM and ODM

Finally, 36 patients with YDM (23.6 ± 4.8 years) and 36 patients with ODM (54.3 ± 10.0 years) were enrolled. The demographic characteristics and MetS components in YDM and ODM are shown in Table 1. In general, FPG, HgbA1c, and TG in YDM are higher compared to ODM. On the other hand, SBP and HDL-C in ODM were higher. The visfatin and RBP4 levels in YDM and ODM are depicted in Figure 2. Visfatin in YDM is found to be higher than that of ODM. However, no significant difference was noted in the levels of RBP4 between YDM and ODM.

### 3.2. Relationship of Visfatin and RBP4 with Metabolic Syndrome Components in YDM and ODM

After adjustment of age and BMI, the correlation of visfatin and RBP4 with MetS components in YDM and ODM are shown in Table 2, respectively. In YDM, visfatin was not related to any MetS components and RBP4 was only positively related to TG. At the same time, in ODM, visfatin was positively related to TG and negatively related to FPG. RBP4 in ODM was positively related to DBP. After analysis of multivariate linear regression with all MetS components, RBP4 was non-significantly and positively related to TG (β = 0.471, *p* = 0.085) in YDM and DBP (β = 0.699, *p* = 0.053) in ODM. However, visfatin was still positively related to TG (β = 0.578, *p* = 0.004) and negatively related to FPG (β = −0.562, *p* = 0.005) in ODM after multivariate linear regression.

## 4. Discussion

To the best of our knowledge, this is the first paper exploring the visfatin and RBP4 levels between YDM and ODM. We found that the visfatin levels are significantly lower in ODM. However, the same difference is not noted in RBP4. After adjustment of age and BMI, visfatin is not related to any components of MetS in YDM, but it was negatively related to FPG and positively to TG in ODM. As for RBP4, our results showed that it is positively related to TG in YDM and to DBP in ODM.

In the present study, we have found that HbA1c, TG were higher and HDL-C were lower in YDM. These results are in line with other mainstream reports. For instance, Ahmad et al. reported similar findings that younger diabetes (<60 years old) patients had higher plasma glucose and lipid levels [22]. Although the age in the study cohort was older, this study still can be regarded as support to our findings. Another study, done by Weinstock et al., also demonstrated that in YDM (age ranging from 10–17 years old), the prevalence of MetS was 75.8% and it was more common in females (83.1%) than males (62.3%) [23]. The most straightforward explanation to cause these differences might derive from less mature psychological abilities to deal with the stress from the disease in YDM. In the Treatment Options for type 2 Diabetes in Adolescents and Youth (TODAY) trial, Walders-Abramson et al. drew the conclusions that, by using major stressors scores such as oral medication adherence, presence of depressive symptoms and impaired quality of life, exposure to major stressful life events is associated with lower adherence and impaired psychosocial functioning among YDM [24].

It is well-known that, compared with the subcutaneous adipose tissue, visceral adipose tissue contributes more to the risk of atherosclerosis and secretes pro-inflammatory cytokines including visfatin. Interestingly, visfatin is also found to be secreted by a variety of cells [25], including the macrophages infiltrating visceral adiposity. This suggests that visfatin is a multifaceted molecule with diverse roles. For example, it is associated with adipose tissue dysfunction and inflammation [7,26,27]. In our present study, individuals diagnosed with YDM display significantly elevated levels of circulating visfatin compared to those with ODM. This discrepancy may be attributed to the higher BMI found in YDM subjects. Hence, in the case of YDM, obesity could potentially exert a greater influence on the onset of glucose dysregulation. Conversely, for ODM, factors beyond obesity, such as age, significantly impact several aspects of the disease. These factors include a reduction in muscle glucose uptake and an impaired ability to secrete insulin [28]. At the same time, RBP4 is well known as one of the biomarkers for IR and impairment of insulin secretion [29]. Several clinical investigations showed that RBP4 levels and severity of IR simultaneously reduced in subjects after treatment with thiazolidinedione, aerobic exercise, or weight reduction [30,31,32]. In addition, elevated RBP4 induced inflammatory cytokines in macrophage and inhibits insulin signals of adipocytes by c-Jun N-terminal kinase- and toll-like receptor 4-dependent and retinol-independent mechanisms [33]. Moreover, RBP4 activates antigen-presenting cells, leading to adipose tissue inflammation and IR [34]. In our study, RBP4 levels in subjects between YDM and ODM showed no significant difference. It may also suggest that IR is basically similar in patients with YDM and ODM. Accordingly, adipose tissue inflammation accompanied by various inflammatory cytokines contributes to the development of T2DM in patients with either YDM or ODM.

Not surprisingly, visfatin secreted from visceral adipocytes is positively related to TG. Many studies also showed the same relationship between visfatin and TG [35,36], which is consistent with our results. Nevertheless, a similar association is not disclosed in YDM may be due to the age-related protective effect and higher basal metabolic rate in them [37]. At the same time, surprisingly, we found visfatin was negatively related to FPG in the ODM group. This result is against the generally recognized concept that visfatin is related to IR. However, our finding is not alone. Liang et al. explored 38 patients with gestational diabetes mellitus. They also found a similar associated (r = −0.47) [38]. The possible explanation behind this observation might be that visfatin was found to have the characteristic of mimicking insulin and could stimulate glucose uptake in muscle [39]. In addition, visfatin might also promote differentiation and maturation of preadipocytes, further promoting glucose transport, lipogenesis, and accumulation in visceral fat [40]. Again, this relationship could not be found in YDM. To explain this, we hypothesize that young adults might have a more compensative ability for glucose homeostasis, which weakens its relationship.

The roles of RBP4 in T2D have been of interest. Gavi et al. explored the 48 young and 55 elderly health subjects and compared RBP4 levels with MetS components and insulin sensitivity in different age groups [41]. They found RBP4 is related to trunk fat, TG, and LDL in the young group, but the association did not show in the elderly group. In the present study, we only found that it is positively related to TG in the YDM group but not in the ODM group. This discrepancy might be explained by the nature of the study group, ours were diabetic and theirs were healthy. Thus, it is possible that hyperglycemia in YDM attenuates these relationships and further evidence is needed to support this hypothesis.

Several clinical investigations showed that higher RBP4 levels in subjects are related to hypertension [42,43] and associated with endothelial dysfunction [44]. This effect could also be observed in animal models. By lowering RBP4, the eNOS-mediated vasodilatation RBP4 could be enhanced in knockout mice [45]. These findings are all in line with the results in the present study.

There are some limitations in our study. First, the number of subjects in our study is relatively small because the prevalence of patients with YDM (onset age < 22 years) is lower than in ODM. According to the Taiwan Nationwide Health Insurance database, the prevalence of YDM (age of 18–22) was less than 0.5% in Taiwan [46], which increased the difficulty of enrollment in our study. A large cohort study may be needed in the future. Second, although adipokines are reported to be related to IR, this association could be affected by many factors such as diet, exercise, and smoking status. In addition, several adipokines secreting from adipocytes may have a crosstalk influence on each other. We did not have this information in the present study and, thus, could not adjust these confounding factors. This is important particularly when the relationships are weak. Third, we did not explore the levels of RBP4 and visfatin in age-match healthy subjects as a control group. Several studies showed visfatin and RBP4 levels increased in patients with T2D [8,9] and may contribute to the pathophysiology of the development of T2D.

## 5. Conclusions

In conclusion, a higher visfatin level was noted in patients with YDM, and it was positively related to TG and negatively to FPG only in ODM. At the same time, RBP4 was positively associated with TG in YDM and DBP in ODM. The roles of both adipokines are important and need further study to clarify.

## Figures and Tables

**Figure 1 medicina-59-01278-f001:**
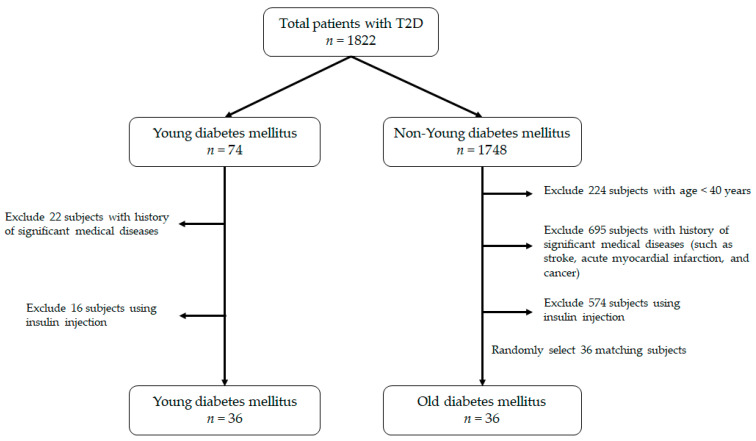
The flowchart of the process on patient selection.

**Figure 2 medicina-59-01278-f002:**
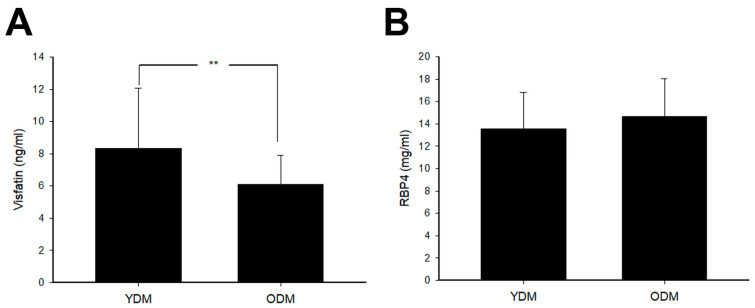
The circulating (**A**) visfatin and (**B**) retinol binding protein 4 (RBP4) levels in subjects with young onset type 2 diabetes mellitus (YDM) and old diabetes mellitus (ODM). (**A**) Visfatin levels in YDM were significantly higher than those in ODM. (** *p* < 0.01). (**B**) RBP4 levels showed no significant difference between YDM and ODM.

**Table 1 medicina-59-01278-t001:** General characteristics and metabolic syndrome components between young diabetes mellitus (YDM) and old diabetes mellitus (ODM).

	YDM	ODM
*n*	36	36
Sex (men/women)	27/9	20/16
Age (year)	23.6 ± 4.8	54.1 ± 10.1
BMI (Kg/m^2^)	26.5 ± 4.7	24.7 ± 2.9
WC (cm)	85.9 ± 14.3	85.4 ± 7.1
SBP (mmHg)	117.6 ± 13.8	125.0 ± 14.3 *
DBP (mmHg)	77.9 ± 9.7	82.9 ± 11.2
FPG (mmol/l)	9.96 ± 3.37	7.85 ± 2.41 *
HgbA1c (%)	10.6 ± 2.8	8.6 ± 1.9 *
TG (mmol/l)	2.15 ± 1.35	1.60 ± 0.76 *
HDL-C (mmol/l)	0.97 ± 0.44	1.38 ± 0.32 *
Medications		
Sulfonylurea (*n* (%))	22 (61.1)	24 (66.7)
Metformin (*n* (%))	31 (86.1)	32 (88.9)
Troglitazone (*n* (%))	8 (22.2)	7 (19.4)

Data shown as mean ± standard deviation; BMI: body mass index; WC: waist circumference; FPG: fasting plasma glucose; HgbA1c: glycated hemoglobin A1c; SBP: systolic blood pressure; DBP: diastolic blood pressure; TG: triglyceride, HDL-C: high density lipoprotein-cholesterol. * *p* < 0.05 as compared with YDM.

**Table 2 medicina-59-01278-t002:** The relationships in visfatin and RBP4 levels with metabolic syndrome components in subjects with YDM or ODM after adjusting for age and BMI.

	YDM		ODM	
	Visfatin	RBP4	Visfatin	RBP4
WC	−0.241	0.181	0.195	0.140
SBP	−0.167	0.387	0.038	0.370
DBP	0.320	−0.169	0.055	0.480 *
FPG	−0.223	0.062	−0.568 *	0.320
TG	−0.255	0.539 *	0.410 *	0.120
HDL-C	−0.081	−0.229	0.120	0.235

* *p* < 0.05; Abbreviations as footnotes in Table 1.

## Data Availability

Data available on request due to privacy/ethical restrictions.

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
