# Peer review of "Visfatin and Retinol Binding Protein-4 in Young-Onset Type 2 Diabetes Mellitus"

_medicina, 2023, doi:10.3390/medicina59071278_

Round 1
Reviewer 1 Report
This research compares the circulating visfatin and RBP4 levels between young-onset type 2 diabetes mellitus (YDM) and late-onset type 2 diabetes mellitus (ODM) and their relationships with metabolic syndrome components in good explanation. It is better to mention what oral hypoglycemic agents are used in the material and methods. In the discussion part, it is better to use the latest article about visfatin and RBP4 and its relation to type 2 diabetes since some studies have been done about this topic.
Author Response
We would like to express our gratitude to the reviewer for taking the time to review our article and providing valuable feedback. We have carefully considered the reviewer's recommendations and opinions. We appreciate the reviewer's efforts to help us improve the quality of our article. We would like to take the reviewer's questions and comments into account and have made the necessary corrections and revisions to the article. We have answered the reviewer's questions point to point and made sure that our article is now more accurate and informative.
This research compares the circulating visfatin and RBP4 levels between young-onset type 2 diabetes mellitus (YDM) and late-onset type 2 diabetes mellitus (ODM) and their relationships with metabolic syndrome components in good explanation. It is better to mention what oral hypoglycemic agents are used in the material and methods.
ANS: We appreciate the suggestions from the reviewers. Given the rare occurrence of YDM, we conducted patient recruitment between 2009 and 2012. This timeline was prior to the introduction of DPP4-IV, SGLT-2i, and GLP-1 medications. To lessen the effects of various medications, we intentionally chose not to incorporate these three drugs even after their market availability. Additionally, to preserve consistency, we deliberately excluded subjects who received insulin injections. The distribution of the administered medications is detailed in the table 1.
In the discussion part, it is better to use the latest article about visfatin and RBP4 and its relation to type 2 diabetes since some studies have been done about this topic.
ANS: We deeply appreciate the insightful recommendations provided by the reviewer, which have greatly contributed to the improvement of our work. We have updated our manuscript with the most recent references about visfatin and RBP4.
Once again, we would like to thank the reviewer for their time and effort in reviewing our article. Their comments and suggestions have been immensely helpful in improving the quality of our work. We hope that our revised article meets the high standards of your esteemed journal.
Sincerely,
Chung-Ze Wu M.D.;Ph.D.
Division of Endocrinology and Metabolism, Department of Internal Medicine, School of Medicine, College of Medicine, Taipei Medical University, Taiwan.
Reviewer 2 Report
Introduction:
The author's should briefly discuss the role of visfatin and RBP-4 in MetS, diabetes, NAFLD, and cardiovascular disease.
Methods:
According to the ADA criteria for the diagnosis of diabetes, the authors mentioned that they chose just one out of the three criteria. Please clarify which specific criterion was chosen.
Please provide more details about the endocrine disorders of the patients that were excluded from the study.
Specify which diabetes medications were excluded from the study.
Over 60% of the literature cited is outdated and lacks new evidence. It is recommended that the authors update the literature and references accordingly.
Results:
In the baseline characteristics, it would be helpful to know which diabetic medications the subjects were taking. Were any of them on TZDs and/or SGLT2 inhibitors?
It would also be helpful to know the number of subjects in each category who were on GLP1-RA.
Discussion:
In the third paragraph of the discussion section, it is recommended to rephrase the sentence from [Ref 16] to improve comprehension
To enhance the overall clarity and readability, a minor grammatical revision is suggested
Author Response
We would like to express our gratitude to the reviewer for taking the time to review our article and providing valuable feedback. We have carefully considered the reviewer's recommendations and opinions. We appreciate the reviewer's efforts to help us improve the quality of our article. We would like to take the reviewer's questions and comments into account and have made the necessary corrections and revisions to the article. We have answered the reviewer's questions point to point and made sure that our article is now more accurate and informative.
Introduction:
The author's should briefly discuss the role of visfatin and RBP-4 in MetS, diabetes, NAFLD, and cardiovascular disease.
ANS: We sincerely appreciate your engagement with our work and have carefully considered your suggestions. In response, we have made significant revisions and included additional information regarding the relevance of visfatin and RBP4 to cardiovascular diseases and type 2 diabetes-related diseases in Introduction Section. We sincerely hope that these additions will provide a more comprehensive view of visfatin and RBP4's roles in type 2 diabetes mellitus and that readers will benefit from a clearer understanding of the subject matter.
Methods:
According to the ADA criteria for the diagnosis of diabetes, the authors mentioned that they chose just one out of the three criteria. Please clarify which specific criterion was chosen.
ANS: In our present study, we did use fasting plasma glucose as the diagnostic criteria for type 2 diabetes. Please refer to the 'Subjects' subsection within the 'Methods' section, where it states, “The diagnostic criteria of T2D were based on the ADA guideline (fasting plasma glucose ≧ 7 mmol/dL). All subjects are being treated with only oral hypoglycemic agents at the time of enrollment.”
Please provide more details about the endocrine disorders of the patients that were excluded from the study.
ANS: We have furnished a flowchart to guide patient selection. Please refer to the Figure 1.
Specify which diabetes medications were excluded from the study.
ANS: Given the rare occurrence of YDM, we conducted patient recruitment between 2009 and 2012. This timeline was prior to the introduction of DPP4-IV, SGLT-2i, and GLP-1 medications. To lessen the effects of various medications, we intentionally chose not to incorporate these three drugs even after their market availability. Additionally, to preserve consistency, we deliberately excluded subjects who received insulin injections. The distribution of the administered medications is detailed in the Table 1
Over 60% of the literature cited is outdated and lacks new evidence. It is recommended that the authors update the literature and references accordingly.
ANS: We deeply appreciate the insightful recommendations provided by the reviewer, which have greatly contributed to the improvement of our work. We have updated our manuscript with the most recent references about visfatin and RBP4.
Results:
In the baseline characteristics, it would be helpful to know which diabetic medications the subjects were taking. Were any of them on TZDs and/or SGLT2 inhibitors? It would also be helpful to know the number of subjects in each category who were on GLP1-RA.
ANS: Given the rare occurrence of YDM, we conducted patient recruitment between 2009 and 2012. This timeline was prior to the introduction of DPP4-IV, SGLT-2i, and GLP-1 medications. To lessen the effects of various medications, we intentionally chose not to incorporate these three drugs even after their market availability. Additionally, to preserve consistency, we deliberately excluded subjects who received insulin injections. The distribution of the administered medications is detailed in the Table 1.
Discussion:
In the third paragraph of the discussion section, it is recommended to rephrase the sentence from [Ref 16] to improve comprehension
ANS: Thanks reviewer’s recommendation. We have rephrased the paragraph as follows, hoping that our explanation will make our description clearer.
“In our present study, individuals diagnosed with YDM display significantly elevated levels of circulating visfatin compared to those with ODM. This discrepancy may be attributed to the higher BMI found in YDM subjects. Hence, in the case of YDM, obesity could potentially exert a greater influence on the onset of glucose dysregulation. Conversely, for ODM, factors beyond obesity, such as age, significantly impact several aspects of the disease. These factors include a reduction in muscle glucose uptake and an impaired ability to secrete insulin.”
Once again, we would like to thank the reviewer for their time and effort in reviewing our article. Their comments and suggestions have been immensely helpful in improving the quality of our work. We hope that our revised article meets the high standards of your esteemed journal.
Sincerely,
Chung-Ze Wu M.D.;Ph.D.
Division of Endocrinology and Metabolism, Department of Internal Medicine, School of Medicine, College of Medicine, Taipei Medical University, Taiwan.